# CAR-NK Cells Generated with mRNA-LNPs Kill Tumor Target Cells In Vitro and In Vivo

**DOI:** 10.3390/ijms241713364

**Published:** 2023-08-29

**Authors:** Vita Golubovskaya, John Sienkiewicz, Jinying Sun, Shiming Zhang, Yanwei Huang, Hua Zhou, Hizkia Harto, Shirley Xu, Robert Berahovich, Lijun Wu

**Affiliations:** 1Promab Biotechnologies, 2600 Hilltop Drive, Richmond, CA 94806, USA; john.sienkiewicz@promab.com (J.S.); hua.zhou@promab.com (H.Z.); hizkia.harto@promab.com (H.H.); robert.berahovich@promab.com (R.B.); 2Laboratory for Critical Quality Attributes of Cell Therapy Products, Forevertek Biotechnology, Janshan Road, Changsha Hi-Tech Industrial Development Zone, Changsha 410205, China

**Keywords:** natural killer cells, chimeric antigen receptor, lipid nanoparticles, mRNA, leukemia, multiple myeloma, tumor, immunotherapy

## Abstract

Natural killer (NK) cells are cytotoxic lymphocytes that are critical for the innate immune system. Engineering NK cells with chimeric antigen receptors (CARs) allows CAR-NK cells to target tumor antigens more effectively. In this report, we present novel CAR mRNA-LNP (lipid nanoparticle) technology to effectively transfect NK cells expanded from primary PBMCs and to generate functional CAR-NK cells. CD19-CAR mRNA and BCMA-CAR mRNA were embedded into LNPs that resulted in 78% and 95% CAR expression in NK cells, respectively. BCMA-CAR-NK cells after transfection with CAR mRNA-LNPs killed multiple myeloma RPMI8226 and MM1S cells and secreted IFN-gamma and Granzyme B in a dose-dependent manner in vitro. In addition, CD19-CAR-NK cells generated with CAR mRNA-LNPs killed Daudi and Nalm-6 cells and secreted IFN-gamma and Granzyme B in a dose-dependent manner. Both BCMA-CAR-NK and CD19-CAR-NK cells showed significantly higher cytotoxicity, IFN-gamma, and Granzyme B secretion compared with normal NK cells. Moreover, CD19-CAR-NK cells significantly blocked Nalm-6 tumor growth in vivo. Thus, non-viral delivery of CAR mRNA-LNPs can be used to generate functional CAR-NK cells with high anti-tumor activity.

## 1. Introduction

CAR-T cells were recently approved by the FDA to treat hematological cancers (leukemia, lymphoma, and multiple myeloma) and demonstrated promising results [1,2,3,4]. CAR-T cell therapy made impressive advancements in the field of cancer immunotherapy but has several limitations. Cytokine release storm (CRS), neurotoxicity, and challenges to target solid tumors are among some potential drawbacks for CAR-T cell therapy [5,6]. CAR-NK cell therapy is another type of promising treatment for cancers [5,6,7]. One of the advantages of NK cells is a low risk of graft-versus-host disease (GVHD) and low toxicity [8,9]. NK cells are also good candidates for allogeneic cell therapy as they show independence to HLA-TCR recognition signaling of T cells [10].

CAR-NK cells were used in preclinical studies against B-cell malignancies [7,11,12]; multiple myeloma [13,14]; and solid tumors such as glioblastoma [15,16], breast [17,18], and ovarian cancers [19]. There are several clinical trials ongoing with CAR-NK cells against hematological and solid tumors [5] that support the use of CAR-NK cells for the treatment of various cancers.

CAR-NK cells can be made with viral and non-viral delivery of CAR. The use of viral vectors is associated with high cost, regulatory requirements, and some safety concerns [20]. There are several non-viral methods such as RNA electroporation and DNA transfection, but these methods have limitations due to their low efficiency of transfection to NK cells [20]. Developing a non-viral method for the generation of CAR-NK cells has advantages for manufacturing due to lower cost, easier preparation compared with the viral transduction route, and convenience for the development of allogeneic off-the-shelf CAR-NK cells.

Recently, lipid nanoparticles (LNPs) were successfully used by Moderna and Pfizer for the delivery of COVID-19 mRNA vaccines [21]. The nanoparticle-based mRNA delivery demonstrated many advantages, such as high stability, bioavailability, solubility, and low toxicity [22]. In this report, we developed a non-viral delivery of CAR mRNA to NK cells expanded from primary PBMCs using mRNA-LNP technology [21,22]. NK cells were expanded from primary PBMCs using the method with K562 feeder cells expressing 4-1BB ligand and membrane-bound IL-21 (K562-4-1BBL/mbIL-21), which stimulate NK cell activity and enhance in vitro expansion [23]. We demonstrated high expansion of NK cells (>8000-fold) and high efficiency of CAR mRNA-LNP transfection resulting in >75% CAR-positive NK cells. BCMA-CAR-NK effectively killed multiple myeloma cells and secreted a high level of IFN-gamma and Granzyme B. CD19-CAR-NK cells effectively killed leukemia cancer cells and secreted high levels of IFN-gamma and Granzyme B in vitro. Moreover, CD19-CAR-NK cells significantly blocked Nalm-6 leukemia tumor growth in vivo. Thus, this report, for the first time, demonstrates that CAR-NK cells generated using the mRNA-LNP technology platform are highly functional in vitro and in vivo and can provide a basis for future preclinical and clinical applications.

## 2. Results

### 2.1. Effective Expansion of NK Cells and Transfection with GFP-mRNA-LNP

For the expansion of NK cells, NK cells were initially isolated from PBMC using a Miltenyi NK Cell Isolation Kit through CD56-negative selection (Section 4). The isolated CD56-positive NK cells were combined with K562 feeder cells expressing 41BB ligand and membrane-bound IL-21, which were pre-treated with mitomycin C (Figure 1A). The average NK cell expansion with a K562-4-1BBL/mbIL-21 feeder cell line was 8878-fold for three different donors (Figure 1B). The expanded NK cells were 98% CD56-positive and CD3-negative (Figure 1C).

To check the transfection efficiency of NK cells with mRNA-LNPs, we prepared GFP-mRNA-LNPs using the PreciGenome FlexS microfluidic NanoSystem and transfected expanded NK cells (Section 4). GFP expression was 98% 16 h after transfection (Figure 1D). The NK cells were frozen in a D10 cryopreservation medium 16 h post-transfection. The frozen GFP mRNA-LNP-transfected NK cells maintained this level of expression immediately post-thaw and showed a high level of GFP expression (94.5%) up to 72 h post-thaw (Figure 1D). Thus, NK cells were efficiently expanded using K562-41BBL/mbIL-21 feeder cells and effectively transfected with GFP mRNA-LNPs.

### 2.2. CAR-NK Cells Express a High Level of CAR after Transfection with mRNA-LNP

To generate CAR-NK cells, CD19-Flag-CAR mRNA and humanized BCMA-CAR mRNA were embedded into LNPs and used to transfect NK cells, which were about 500-fold expanded. The average size (hydrodynamic diameter) of CD19-CAR mRNA-LNPs was 92.3 nm, with a polydispersity index (PDI) of 0.12. The average size of BCMA-CAR mRNA-LNPs was 96.9 nm, with a PDI of 0.13 (Section 4). CD19-CAR mRNA-LNP transfection resulted in a 78.8% CAR-positive cell population (Figure 2A), and BCMA-CAR mRNA-LNP transfection resulted in a 95.3% CAR-positive cell population (Figure 2B). CD19-CAR-NK cells as well as NK cells were CD3-negative and expressed high levels of CD56 (99%), activation receptor CD337 (NKp30) (99%), and CD16 (95%) (Figure 2C). The same high expression of NK cell markers was observed for BCMA-CAR-NK cells generated with BCMA-CAR mRNA-LNP transfection (not shown). Thus, CAR mRNA-LNP transfection of NK cells expanded from PBMC generated CAR-NK cells with high levels of CAR expression and phenotypic markers of NK cells.

### 2.3. BCMA-CAR-NK Cells Kill Multiple Myeloma Cancer Cell Lines and Secrete Higher Levels of IFN-Gamma and Granzyme B Than NK Cells

To test the functional activity of CAR-NK cells, first BCMA-CAR-NK and non-transfected NK cells were used in a real-time cytotoxicity assay (RTCA) with multiple myeloma RPMI8226 cells at different E:T (effector to target cell) ratios (Figure 3A). BCMA-CAR-NK killed RPMI-8226 cells in a dose-dependent manner more effectively than NK cells (Figure 3A). A similar result was obtained with another multiple myeloma cell line, MM1S (Figure 3B). BCMA-CAR-NK cells killed target cells more effectively than NK cells (Figure 3B).

Supernatant was collected after RTCA killing assay to test the IFN-gamma secretion levels by NK and CAR-NK cells using ELISA (Figure 3C,D). BCMA-CAR-NK cells secreted IFN-gamma in a dose-dependent manner with significantly higher levels compared with non-transfected NK cells against target RPMI8226 cells (Figure 3C) and MM1S cells (Figure 3D). BCMA-CAR-NK cells also secreted Granzyme B in a dose-dependent manner against target RPMI8226 cells (Figure 3E) and MM1S cells (Figure 3F). Thus, transfection of NK cells with BCMA-CAR mRNA-LNPs generates functional BCMA-CAR-NK cells with high killing activity and IFN-gamma and Granzyme B secretion.

### 2.4. CD19-CAR-NK Cells Kill Leukemia Cells and Secrete Higher Levels of IFN-Gamma and Granzyme B Than NK Cells

CD19-CAR-NK cells generated by CD19-CAR-mRNA-LNP transfection were functionally tested using RTCA, IFN-gamma and Granzyme B ELISA, and a luciferase cytotoxicity assay (Figure 4). CD19-CAR-NK cells killed Daudi target cells more effectively than NK cells in the RTCA (Figure 4A). CD19-CAR-NK cells secreted IFN-gamma (Figure 4B) and Granzyme B (Figure 4C) at significantly higher levels than NK cells. Similar results were obtained in the luciferase killing assay with Nalm-6-luciferase target cells (Figure 4D). CD19-CAR-NK cells killed Nalm-6-luciferase-positive cells more effectively than NK cells (Figure 4D). CD19-CAR-NK secreted IFN-gamma (Figure 4E) and Granzyme B (Figure 4F) in a dose-dependent manner at significantly higher levels compared with non-transfected NK cells. Thus, transfection of NK cells with CD19-CAR mRNA-LNPs generates functional CD19-CAR-NK cells against leukemia tumor cells. 

### 2.5. CD19-CAR-NK Significantly Decreases Nalm-6 Tumor Growth In Vivo

To test the in vivo activity of human CAR-NK cells, we used Nalm-6-luciferase leukemia cancer cells in the immunodeficient NSG mouse model. On day 0, 1 × 10^5^ Nalm-6-luc+ cells were intravenously injected into NSG mice. Then, 5 × 10^6^ frozen CD19-CAR-NK and NK cells were injected into mice on days 1, 3, 6, and 8, and imaging was carried out on days 6, 9, 12, and 15 (Figure 5A). Both NK and CD19-CAR-NK blocked Nalm-6-luciferase+ tumor growth (Figure 5B). Importantly, CD19-CAR-NK cells blocked Nalm-6-luciferase tumor growth significantly more than NK cells (*p* < 0.00002) by day 15 (Figure 5C). Thus, CD19-CAR-NK cells generated by CD19-CAR mRNA-LNP transfection of NK cells showed high efficacy against Nalm-6 leukemia tumors in vivo.

## 3. Discussion

This report describes the transfection of expanded NK cells using CD19-CAR mRNA-LNPs and BCMA-CAR mRNA-LNPs resulting in high transfection efficiency, with CAR-NK cells showing >78% CAR expression. Humanized BCMA-CAR-NK cells generated with BCMA-CAR mRNA-LNP killed multiple myeloma cell lines RPMI8226 and MM1S in a dose-dependent manner more efficiently than non-transfected NK cells. BCMA-CAR-NK cells also secreted significantly more IFN-gamma and Granzyme B than NK cells against multiple myeloma target cells. CD19-CAR-NK cells generated using CD19-CAR mRNA-LNPs killed Daudi and Nalm-6 leukemia cell lines in a dose-dependent manner and secreted IFN-gamma and Granzyme B at significantly higher levels than NK cells. In addition, CD19-CAR-NK cells significantly decreased Nalm-6-tumor growth in vivo.

We observed that transfection using CAR mRNA-LNPs resulted in different levels of CAR expressions in NK cells. This may be explained by differences of the ScFv in CAR. For example, CD19-CAR expression was 78.8% and BCMA-CAR expression was 95.3%. The expression of CAR can be increased by codon optimization of DNA templates to generate a uridine-depleted mRNA sequence.

This report demonstrates efficient expansion of NK cells isolated from primary PBMCs using a K562 feeder cell line expressing 4-1BB ligand and membrane-bound IL-21. The average expansion of NK cells from three different donors was greater than 8000-fold at day 18 post-activation. This method can be used in a clinic as described by [23,24] for the manufacturing of allogenic CAR-NK cells. The authors described quality assurance tests needed for manufacturing NK cells such as the level of GFP-positive irradiated K562 feeder cells, endotoxin level, sterility, mycoplasma level, purity, viability, and functional activity of clinical-grade NK cells [24]. Using gas-permeable flasks (G-Rex) allows for large-scale expansion of NK cells, which are usually present at low levels in PBMCs (in the range of 5–20%). Expansion of NK cells directly from PBMCs can be prone to difficulties. G-Rex flasks with gas exchange directly occurring across the bottom of the cell culture allows for increased volumes of medium with less cell manipulation needed, increases the rate of cell expansion, and increases cell viability [24]. The authors used a K562-41BBL/mb IL-15 feeder cell line with the G-Rex system, while this report demonstrates high and comparable expansion of NK cells using a K562 cell line expressing 4-1BBL and membrane-bound IL-21. We evaluated our expanded NK cells using the above-described manufacturing parameters [24] and found an almost completely negative level of irradiated GFP-positive K562 feeder cells in the final product, high purity, sterility, and functional activity of expanded NK cells. In addition, we optimized the expansion of NK cells with human serum in the culture medium and generated CAR-NK cells with CAR mRNA-LNP transfection, and we found that CAR-NK cells were functional against target cancer cells (not shown). Thus, expanded NK cells and CAR-NK cells generated with mRNA-LNP transfection can be used for future pre-clinical and clinical development of allogeneic off-the-shelf cell therapies.

The use of mRNA-LNPs is a non-viral technology that allows for easier and cost-effective manufacturing and produces a high number of CAR-positive NK cells. The transient expression of CAR can have advantages in several applications. For example, CD19-CAR-T cells were recently used for the treatment of systemic lupus erythematosus [25]. Thus, treating autoimmune diseases can be tested in future with CAR-NK cells generated with CAR mRNA-LNP transfection, where shorter treatment with transiently transfected CAR-NK can be advantageous due to less exposure time. Another report recently showed that two infusions of IL-15-CD19-CAR-NK cells were more effective than a single infusion of CAR-NK cells, suggesting that multiple injections of CAR-NK cells may be more effective against tumor cells [26]. These results correspond well to the present study that used four injections of CAR-NK cells showing high efficacy in a mouse leukemia Nalm-6 tumor model. CD19-CAR-NK cells had significantly higher efficacy compared with NK cells in blocking Nalm-6 tumor growth.

Another potential application of CAR-NK cells generated using CAR-mRNA-LNPs is their use in treating cardiac diseases similarly to CAR-T cells, which can treat cardiac fibrosis [27,28]. CAR-T cells have been used in a pre-clinical mouse study to target fibrosis using the fibroblast activation protein (FAP) [28]. Thus, CAR-NK applications can be expanded to different diseases where non-viral transient transfection of CAR mRNA-LNP is advantageous due to lower toxicity and lower side effects such as cytokine release syndrome (CRS) associated with CAR-T cell therapies.

CAR-NK cells can also be used in combination with mRNA-LNPs encoding for different cell signaling players, which can enhance the expansion of NK cells and decrease their exhaustion, such as checkpoint inhibitors (PD-L1, PD-1, LAG-3, TIGIT, and TIM-3) and other immunomodulators such as cytokines, chemokines, receptor ligands (IL-12, IL-15, IL-2, Il-18, IL-21, FLT-3, and 4-1BBL), and others. Future studies of these molecular signaling players in combination with CAR-NK cells can be critical for the development of anti-cancer immunotherapies.

## 4. Materials and Methods

### 4.1. Cells

K562, Daudi, Nalm-6, MM1S, and RPMI-8226 cell lines were purchased from ATCC and were cultured in an RPMI-1640 medium supplemented with 10% FBS and penicillin/streptomycin. 293FT cells were obtained from AlStem (Richmond, CA, USA) and were cultured in Dulbecco’s modified eagle’s medium (DMEM) with 10% FBS and penicillin/streptomycin. A Nalm-6-EGFP-luciferase cell line was generated after transducing Nalm-6 cells with luciferase-EGFP lentivirus. K562-41BBL/mbIL21 (membrane-bound) feeder cells were obtained after transduction of K562 cells with lentivirus containing 4-1BBL and IL-21 membrane-bound coding sequences. Human peripheral blood mononuclear cells (PBMCs) were isolated from whole blood obtained from the Stanford Hospital Blood Center, according to the IRB-approved protocol (#13942). PBMCs were isolated by standard density sedimentation over Ficoll–Paque (GE Healthcare, Chicago, IL, USA) and cryopreserved for later use. All cell lines were cultured in a 5% CO_2_ humidified incubator.

### 4.2. Antibodies

Goat anti-mouse IgG, F(ab′)_2_ fragment-specific antibodies were obtained from Jackson Immunoresearch. PE-anti-DYKDDDDK (Flag) tag, PE-streptavidin, and 7-AAD viability staining solution were obtained from BioLegend. 4-1BB ligand and IL-21 antibodies were from BioLegend (San Diego, CA, USA). mIgG1 isotype control; mIgG2a isotype control; and CD3, CD16, and CD56 antibodies were obtained from BioLegend (San Diego, CA, USA). Anti-human CD337 (NKp30) and clone 7E4E10 were obtained from Promab Biotechnologies (Richmond, CA, USA).

### 4.3. Lentivirus Generation

EGFP, luciferase, and 4-1BBL/mb IL-21 lentiviruses were generated using HEK-293FT cells as described in [29]. The lentiviruses were used for the transduction of different cell lines, and protein expression was verified by FACS or by luciferase assay.

### 4.4. NK Cell Isolation and Expansion

NK cells were isolated from PBMCs using an NK Cell Isolation Kit, Human (Miltenyi Biotec) according to the manufacturer’s protocol. The isolated NK cells were expanded by culturing with K562-41-BBL/mb IL-21 feeder cells, which were pre-treated with mitomycin C (Sigma-Aldrich, St Louis, MO, USA). Cells were cultured in non-treated cell culture flasks or in gas-permeable static cell culture flasks (G-Rex) (Wilson–Wolf) [24]. The medium for expansion was AIM-V supplemented with 10% FBS, IL-2 (10 ng/mL), and IL-15 (5 ng/mL). NK cells were frozen using a NutriFreez D10 Cryopreservation Medium (Satorius, Fremont, CA, USA).

### 4.5. mRNA In Vitro Transcription

The mRNA was transcribed in vitro using a HiScribe T7 mRNA Kit with a CleanCap Reagent AG (NEB). The GFP coding sequence was inserted into a DNA template vector with a T7AG promoter in front, 5′UTR and 3′UTR flanking the open-reading frame of the codon sequence, and a 152 poly-A tail after the stop codon. For CD19-CAR mRNA, a CD19 scFv (FMC63) Flag tag-CD28-CD3 sequence [30] was used for inserting into the above vector. For BCMA-CAR mRNA, humanized BCMA scFv-41BB-CD3 CAR was inserted into the DNA template vector [31]. In brief, a DNA template, 0.5 × T7 CleanCap Reagent AG Reaction Buffer, 5 mM of ATP, CTP, pseudo-UTP, and GTP were added to 4 mM of CleanCapAG and T7 polymerase. All combined reagents were mixed and incubated for 2 h at 37 °C. After DNAse I treatment for 15 min at 37 °C, the mRNA was purified with a Monarch RNA Cleanup Kit (NEB) according to the manufacturer’s protocol. mRNA was checked for their correct molecular weight by running on an agarose gel with a molecular weight ladder, and the concentration of mRNA was determined using a spectrophotometer (Nanodrop, ThermoFisher Scientific), Waltham, MA, USA.

### 4.6. mRNA-LNP Generation and Transfection of NK Cells

To generate an mRNA–LNP complex, an aqueous solution of mRNA in 100 mM sodium acetate (pH 4.0) was combined with a lipid mix containing SM-102, DSPC, cholesterol, and DMG-PEG2000 (at a molar % ratio of 50:10:38.5:1.5, respectively, dissolved in ethanol) at a flow rate ratio 4:1 (aqueous: organic phase) using PreciGenome Flex S System (San Jose, CA, USA). The mRNA-LNPs were purified and concentrated using Amicon^®^ Ultra-15 centrifugal filter units (30–100 kDa). The average size and polydispersity index (PDI) were detected using Anton Paar Litesizer 500 System.

CAR mRNA-LNPs (1–2 µg of encapsulated mRNA per 1 × 10^6^ NK cells) were directly added to expanded NK cells. Cells were mixed with mRNA-LNPs and incubated overnight. At 16–24 h after mRNA-LNP transfection, CAR-NK cells were harvested for analysis or were frozen in a NutriFreez D10 Cryopreservation Medium for later use.

### 4.7. FACS

FACS was performed as described in [29,32,33]. In brief, 0.25 million cells were suspended in 100 µL of FACS buffer (PBS containing 2 mM EDTA (pH 8) and 0.5% BSA) and incubated on ice with 1 µL of human serum for 10 min. An appropriate primary antibody was used, and cells were incubated for 30 min at 4 °C. Cells were then washed twice with FACS buffer, and the secondary antibody was added; cells were again incubated for 30 min at 4 °C. Then, cells were rinsed twice with FACS buffer and stained for 10 min with 7-AAD, and FACS analysis was performed on the FACS Calibur (BD Biosciences, San Jose, CA, USA). Viable cell populations were gated by excluding cells that were positive for 7-AAD. Positive expression was then gated by using an appropriate isotype control or by gating against non-transfected cells.

### 4.8. Cytotoxicity Assay

Real-time cell analysis (RTCA), an impedance-based cytotoxicity assay, using CELLigence system (Agilent) was performed with Daudi and multiple myeloma (RPMI8226 and MM1S) cell lines. In brief, 1 × 10^4^ target cells were seeded into a 96-well E-plate covered with CD40-tethering antibody for leukemia cells or CD9-tethering antibody for multiple myeloma cells to attach cells to the plate (Agilent/Acea Biosciences, San Diego, CA, USA). The next day, the medium was removed and replaced with an AIM V-AlbuMAX medium containing 10% FBS. NK and CAR-NK effector cells at various E:T (effector to target) ratios were added in triplicate. The cells were monitored for 24–48 h with the RTCA system, and impedance was plotted over time. Cytotoxicity percent was calculated as follows: (impedance of target cells without effector cells minus impedance of target cells with effector cells)/impedance of target cells without effector cells × 100. Nalm-6 luciferase-EGFP^+^ cells were treated with either NK or CD19-CAR-NK cells at different E:T ratios. The cytotoxicity was quantified by luciferase assay using the luciferase assay substrate from Steady-Glo Luciferase assay system (Promega). The luciferase-positive Nalm-6 viable cells were normalized to untreated Nalm-6-luc+ cells in duplicates, and the percentage of cytotoxicity was calculated at different E:T ratios.

### 4.9. IFN-Gamma and Granzyme B Secretion Assay by ELISA

Nonadherent target cells were cultured with NK or CAR-NK effector cells at different E:T ratios in 96-well plates in triplicate. After 16 h, the top 150 µL of the medium was transferred to V-bottom 96-well plates and centrifuged at 300× *g* for 5 min. The top 120 µL of the supernatant was transferred to a new 96-well plate and analyzed by ELISA for human IFN-gamma or Granzyme B levels using an R&D Systems Human IFN-gamma Quantikine Kit and an R&D Systems Granzyme B Quantikine Kit (Minneapolis, MN, USA) according to the manufacturer’s protocol. The supernatant from RTCAs with adherent target cells was collected after 24–48 h and analyzed as above.

### 4.10. NSG Mouse Model and Imaging

Six-week-old immunodeficient NSG (NOD Scid gamma mouse) mice (Jackson Laboratories, Bar Harbor, ME, USA) were housed in accordance with the Institutional Animal Care and Use Committee (IACUC) (#LUM-001). Each mouse was subcutaneously injected on day 0 with 100 µL of 1 × 10^5^ Nalm-6-luciferase positive cells. Human CAR-NK and NK cells generated by CAR mRNA transfection were frozen 24 h post-transfection. Frozen/thawed 5 × 10^6^ NK or CAR-NK cells were intravenously injected to NSG mice on days 1, 3, 6, and 8. Imaging was performed on days 6, 9, 12, and 15 after luciferin injection using Xenogen Ivis System (Perkin Elmer, Waltham, MA, USA). Quantification of imaging was carried out by measuring bioluminescence (BLI) in photons/s signals.

### 4.11. Statistical Analyses

Comparisons between two groups were performed by Student’s *t*-test. Differences with *p* < 0.05 were considered significant. GraphPad software version 9.5 was used to prepare graphs.

## 5. Conclusions

Thus, this study, for the first time, demonstrates a novel platform to generate CAR-NK cells with mRNA-LNP technology. The NK cells expanded from primary PBMCs expressed high levels of CAR after mRNA-LNP transfection and demonstrated high functional anti-tumor activity in vitro and in vivo. This study provides a basis for future preclinical and clinical studies.

## 6. Patents

The patent application was filed based on these results.

## Figures and Tables

**Figure 1 ijms-24-13364-f001:**
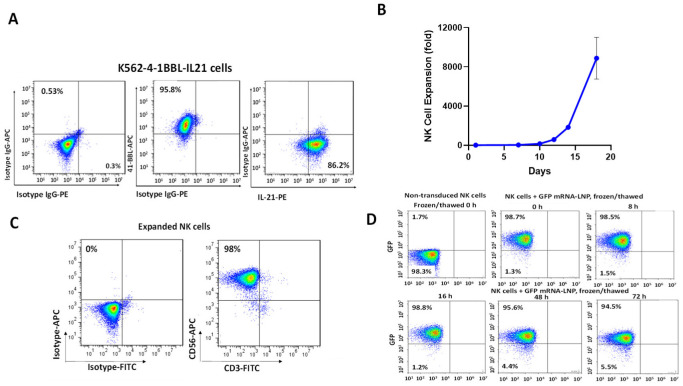
Expansion of NK cells and transfection with mRNP-LNP. (**A**) K562-4-1BB/IL21 feeder cell line used for expansion of NK cells. FACS shows expression of 4-1BBL and mbIL-21 in K562 cells after transduction with lentivirus encoding 4-1BBL and mbIL-21. Expression of 4-1BBL and IL-21 was gated versus isotype control antibodies. (**B**) Expansion of NK cells with K562-41-BBL/IL-21 feeder cell line. Average NK cell expansion using K562-4-1BBL/mbIL-21 feeder cell line is shown for NK cells from three separate NK cell donors. (**C**) Characterization of expanded NK cells. FACS with anti-human CD56 and CD3 antibodies shows >98% CD56-positive and CD3-negative NK cells. Expression of CD56 and CD3 was gated versus isotype control antibodies. (**D**) Transfection of GFP mRNA-LNPs to expanded NK cells results in >94% GFP-positive cells. Expanded NK cells were transfected with GFP mRNA-LNP and frozen 24 h after transfection in D10 cryopreservation medium. After thawing, non-transfected and GFP-mRNA-LNP transfected NK cells were immediately tested for GFP expression by FACS (0 h). The thawed cells were cultured in NK expansion medium, and GFP expression in NK cells (post-thaw) was measured every 24 h over 72 h. GFP-mRNA-LNP transfected NK cells were gated against non-transfected NK cells using the FITC channel. Different colors shows density gradient of stained cells where red—highest density, green—medium and blue—lowest density.

**Figure 2 ijms-24-13364-f002:**
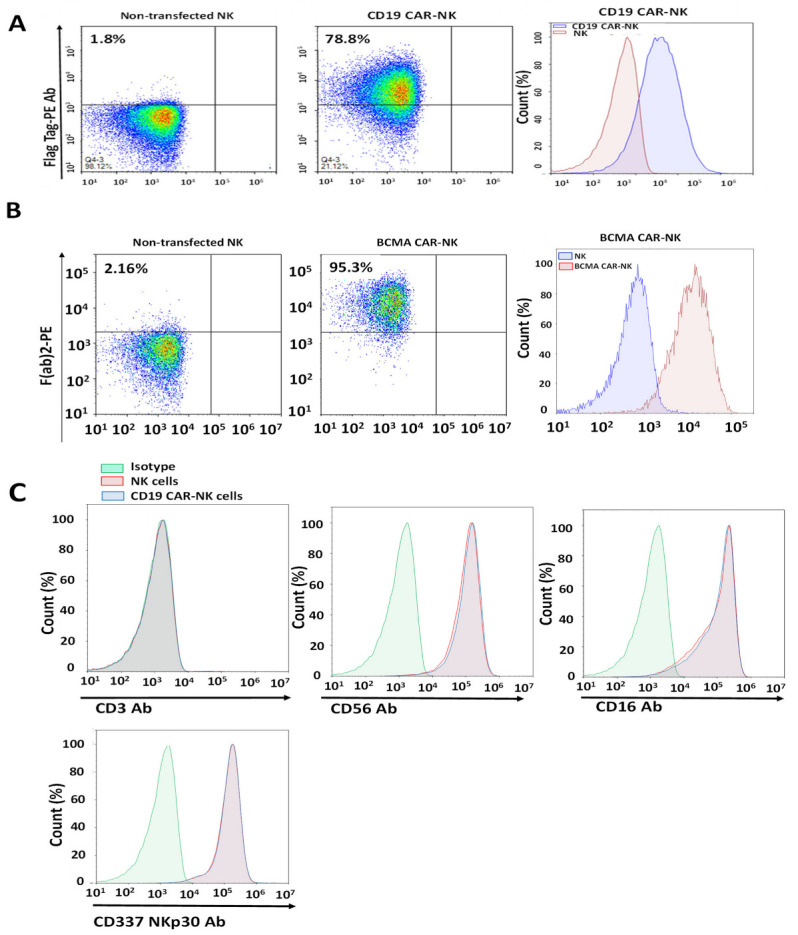
CAR mRNA-LNP transfection to expanded NK cells results in high expression of CAR. CD19-Flag-CAR-NK (**A**) and humanized BCMA-CAR-NK (**B**) cells were analyzed by FACS 16 h post-transfection using CAR mRNA-LNPs. FACS was performed with a Flag tag primary antibody for CD19-CAR-NK and F(ab′)_2_ for humanized BCMA-CAR-NK cells. CD19-CAR-NK and BCMA-CAR-NK cells were gated against non-transfected NK cells. (**C**). The CD19-CAR-NK cells and non-transfected NK cells were CD3-negative and expressed high levels of CD56^+^, CD16^+^, and CD337 (NKp30). CAR-NK cells were analyzed by FACS 16 h after CAR-mRNA-LNP transfection. Positive expression of NK cell markers was gated using isotype controls. Different colors shows density gradient of stained cells where red—highest density, green—medium and blue—lowest density.

**Figure 3 ijms-24-13364-f003:**
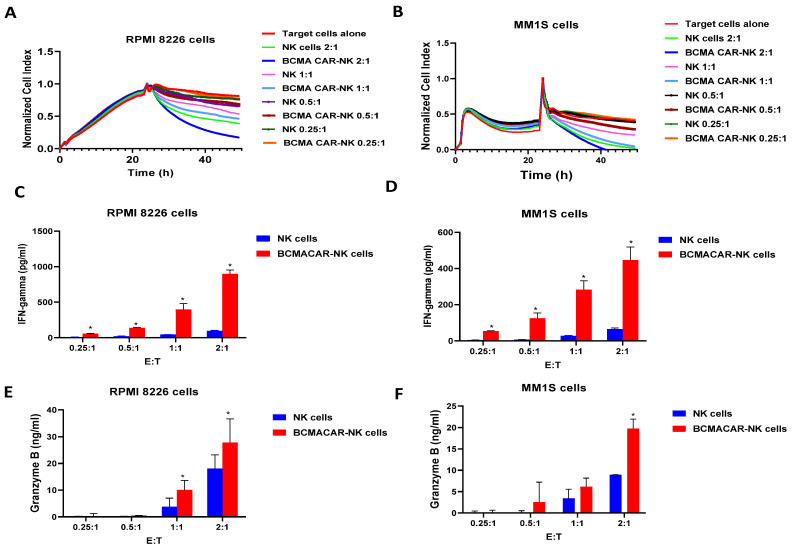
BCMA-CAR-NK cells kill multiple myeloma cell lines and secrete higher levels of IFN-gamma than NK cells in a dose-dependent manner. (**A**). RTCA killing assay with BCMA-CAR-NK effector cells and multiple myeloma RPMI 8226 target cells. (**B**). RTCA with BCMA-CAR-NK effector cells and multiple myeloma MM1S target cells. (**C**). BCMA-CAR-NK cells secrete IFN-gamma in a dose-dependent manner against RPMI8226 multiple myeloma target cells (**C**) and MM1S target cells (**D**), measured by ELISA. BCMA-CAR-NK cells secrete Granzyme B in a dose-dependent manner against RPMI8226 multiple myeloma target cells (**E**) and MM1S target cells (**F**), measured by ELISA. E:T, Effector cells to target cells ratio. Bars show an average of three independent measurements ± standard deviations. * *p* < 0.05, Student’s *t*-test BCMA-CAR-NK cells vs. NK cells, IFN-gamma, and Granzyme B secretion.

**Figure 4 ijms-24-13364-f004:**
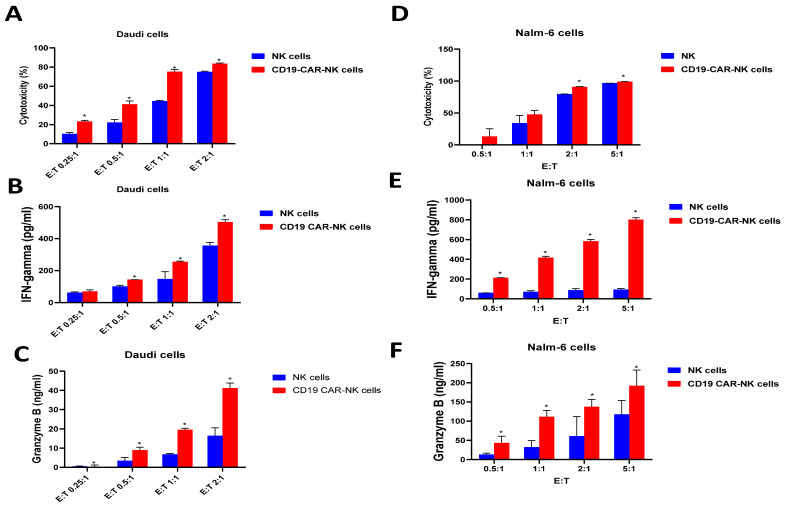
CD19-CAR-NK cells kill leukemia cell lines and secrete higher levels of IFN-gamma and Granzyme B than NK cells in a dose-dependent manner. (**A**). RTCA killing assay with CD19-CAR-NK effector cells and Daudi target cells. Bars show average cytotoxicity of effector cells in RTCA (Section 4) from three independent measurements. * *p* < 0.05 CAR-NK vs. NK, Student’s *t*-test. (**B**). CD19-CAR-NK cells secrete higher levels of IFN-gamma in a dose-dependent manner than NK cells against Daudi cells. * *p* < 0.05 by Student’s *t*-test, CD19-CAR-NK cells vs. NK cells in secretion of IFN-gamma measured by ELISA. (**C**). CD19-CAR-NK cells secrete significantly higher levels of Granzyme B in a dose-dependent manner than NK cells against Daudi cells. * *p* < 0.05 by Student’s *t*-test, CD19-CAR-NK cells vs. NK cells in secretion of Granzyme B measured by ELISA. (**D**). Cytotoxicity assay with Nalm-6-luciferase cells at different E: T ratios by luciferase assay (Section 4). CD19-CAR-NK cell cytotoxicity was significantly higher than NK cells. * *p* < 0.05 CAR-NK cells vs. NK cells by Student’s *t*-test. (**E**). ELISA shows dose-dependent secretion of IFN-gamma by CD19-CAR-NK cells against Nalm-6 target cells. CD19-CAR-NK cells secrete significantly higher levels of IFN-gamma than NK cells against Nalm-6 target cells. * *p* < 0.05 by Student’s *t*-test, CD19-CAR-NK vs. NK cells, IFN-gamma measured by ELISA. (**F**). ELISA shows dose-dependent secretion of Granzyme B by CD19-CAR-NK cells against Nalm-6 target cells. CD19-CAR-NK cells secrete significantly higher levels of Granzyme B than NK cells against Nalm-6 target cells. * *p* < 0.05 by Student’s *t*-test, CD19-CAR-NK vs. NK cells in secretion of Granzyme B measured by ELISA.

**Figure 5 ijms-24-13364-f005:**
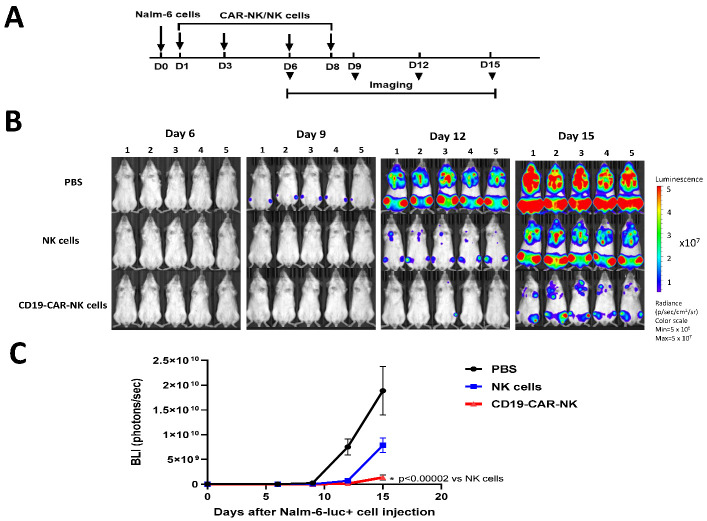
CD19-CAR-NK decreased Nalm-6-luciferase tumor growth significantly more than NK cells using NSG mouse model. (**A**). Scheme of the mouse experiment. 1 × 10^5^ of human leukemia, Nalm-6-luciferase cells were intravenously injected into immunodeficient NSG mice on day 0. Then, 5 × 10^6^ frozen CD19-CAR-NK and NK cells were intravenously injected at days 1, 3, 6, and 8. Imaging was performed with the Xenogen Ivis system on days 6, 9, 12, and 15. N = 5 mice per group. (**B**). CD19-CAR-NK cells significantly decreased Nalm-6 xenograft tumor growth compared with NK cells. The images of mice treated with CAR-NK or NK cells are shown in the Nalm-6 xenograft model. (**C**). Quantification of imaging is shown. Average bioluminescence (photons/s) ± standard deviations is shown on the Y-axis and days on the X-axis. * *p* < 0.00002, CD19-CAR-NK cells vs. NK cells by Student’s *t*-test.

## Data Availability

All data are presented and available inside the manuscript.

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
