# Peer review of "CAR-NK Cells Generated with mRNA-LNPs Kill Tumor Target Cells In Vitro and In Vivo"

_ijms, 2023, doi:10.3390/ijms241713364_

Round 1

Reviewer 1 Report

The manuscript “CAR-NK cells generated with mRNA-LNPs kill tumor target cells in vitro and in vivo” reported effective engineering of NK cells with CAR (CAR-NK cells) via mRNA-LNPs to effectively target multiple myeloma and lymphoma cells. The study is well-designed and functional studies of the engineered cells are extensive. While the results are of significance, the authors should address the following issues.

1.     Since LNPs size and PDI characterizations were performed, it would be interesting to see this data mentioned in the “results” or “Discussion” section.

2.     The discussion section excellently highlights the advantages and potential applications of the engineered cells. It would be interesting to readers if authors could discuss the following points in a statement or two about,

i) CAR mRNA-LNPs transfection in expanded NK cells exhibited differences in % transfected cells. Could this difference be due to LNP size? 

ii) In Nalm-6 cells, there are nearly same levels of IFN-gamma (Fig. 4D) with increased cytotoxicity (Fig. 4C) in NK cells group. Could there be a reason or immune context to this non-linear relationship?

3.     In tumor growth inhibition study, cells were injected at days 1, 3, 6, and 8, while images were taken at days 6, 9, 12, and 15. Is this correct? If authors could check the accuracy of this. Also, if authors could incorporate the time details of procedures in the “Materials and Methods” section. A schematic diagram of the animal study in Fig. 5 will enhance the data representation.

4.     There is a minor typos on line 193. 

Author Response

We would like to thank the reviewer for suggestions which improved the manuscript.

  1. The reviewer suggested including data on LNP characterization (size and PDI) in the Results

We added size and PDI data for CAR-mRNA-LNPs to the Results section 2.1.

The size of mRNA-LNP was similar for different CAR-mRNAs. The average size (hydrodynamic diameter) of CD19(CAR)-mRNA-LNPs was 92.3nm and had a PDI of 0.12. BCMA(CAR)-mRNA-LNPs had a size of 96.9nm; and a PDI of 0.13.

  1. The reviewer suggested including in the Discussion the reason of differences in the % of transfected cells between expanded CD19-CAR-NK and BCMA-CAR-NK cells.

We added to the Discussion that differences can be explained by the nucleotide structure of ScFv in the CAR and that the CAR expression in expanded NK cells can be increased by different codon optimization of DNA template to generate uridine-depleted mRNA or an optimized translated mRNA sequence.

  1. The reviewer asked to explain nearly same levels of IFN-gamma secretion for NK cells with increased cytotoxicity for this group (old Fig 4D).

The level of secreted IFN-gamma by NK cells was low and increased from 60 to 91 pg/ml against Nalm-6 target cells at E:T=0.5:1 to E:T=5:1, respectively (p-value was <0.05) (Fig 4E). To expand on the functional activity of CD19-CAR-NK cells against Nalm-6 cells, we included additional data for CD19-CAR-NK and NK cells showing their Granzyme B secretion, where CD19-CAR-NK cells had significantly higher levels of secreted Granzyme B compared to NK cells when incubated with leukemia target cell lines in (Fig. 4 C, F). Both NK and CAR-NK cells had linear dose-dependent increase of Granzyme B secretion against leukemia cells (Fig. 4 C, F). We also included Granzyme B secretion data for BCMA-CAR-NK cells against target cells (Fig 3 E, F). Multiple players are involved in CAR-NK and NK-mediated cytotoxicity.

  1. The reviewer suggested adding details of the procedure and time points of the imaging for the mouse experiment as well as adding a schematic diagram of the animal study for Figure 5.

We added time points for the imaging shown in Results to the Materials and Methods section and included a schematic diagram (Fig. 5A) for the revised Figure 5.

  1. The reviewer had a comment on a minor typo in line 193.

We corrected the typo.

Reviewer 2 Report

Overall: Vita Golubovskaya et al. develop lipid nanoparticle method applied as non-viral delivery of CAR mRNA to NK cells. This article describes several advantages of this method concerning to the safety, efficacy, and the specific activity. However, these data deserve to be completed.

Comments:

·      Detail the gating strategy concerning to NK cell in Figure 1

·      Can you clarify the positive signal observed on figure 1D at 0h "NK cells + GPF mRNA-LNP, frozen/hawed"

·      On figure 1, two human donor is not enough. It is important to replicate the experiment enough time to support the strength of this innovation.

·      Did you observe some other phenotypical changes on the CAR-NK cells compare NK cells?

·      Did you measure other cytokines produced by the CAR-NK cells in the coculture with target cells?

·      In the figure 5, is it human CAR-NK cells injected in the mice? It is more valuable to used autologous NK cells concerning to the in vivo relevance. The LNP method was applied to mice NK cells? If yes, it is important to show the methodology in mice too.

Author Response

We would like to thank the reviewer for the suggestions which improved the manuscript.

  1. The reviewer suggested adding details on the gating strategy for NK cells in Figure 1.

We added a gating strategy in Figure 1 legends and in Materials and Methods.

  1. The reviewer asked to clarify the positive GFP signal at 0h for GFP mRNA-LNP transfected NK cells in Figure 1D.

We added to the Results section and the Figure legend that NK cells were frozen 24 hours after GFP-mRNA-LNP transfection. After thawing, cells were tested for GFP expression by FACS (0 h timepoint) and measured again at 24h intervals thereafter. We made clearer labels for Figure 1D by adding non-transfected NK cells title on top in the left panel to present clearer for the reader.

  1. The reviewer had a comment that more donors needed to be tested for the Figure 1 data.

For original Figure 1 B, we included the representative expansion curve of same donor from two independent experiments. We replicated high expansion of NK cells using two other donors.  We added revised Figure 1B with expansion on NK cells from three different donors 18-days post-activation. Consistent high expansion of NK cells with K562-4-BB, IL21 feeder cell line was observed in several donors.

  1. The reviewer asked about phenotypic changes of CAR-NK versus NK cells after mRNA-LNP transfection.

We tested phenotypic markers of NK cells: CD56, CD16 and NKp30 (CD337) NK activation receptor. We added a new Figure 2C with FACS data on analysis of phenotypic markers for CAR-NK and NK cells. No significant differences were observed between NK and CAR-NK cells for the phenotypic markers after CAR mRNA-LNP transfection.

  1. The reviewer asked if we tested other secreted proteins in addition to IFN-gamma.

We performed analysis of Granzyme B secretion by CAR-NK versus NK cells against target cells. We added new Figures 3 E and 3F showing Granzyme B secretion by BCMA-CAR-NK cells against multiple myeloma cells, and Figures 4C and 4F showing Granzyme B secretion by CD19-CAR-NK cells against leukemia target cell lines. We found that BCMA-CAR-NK cells secreted significantly higher level of Granzyme B compared to NK cells against target cells.

  1. The reviewer asked if we injected human CAR-NK cells to mice and suggested showing the methodology for the mice experiment.

We injected the human Nalm-6 cancer cell line as well as human CAR-NK and NK cells produced with CAR-mRNA-LNP transfection using immunodeficient NSG xenograft mouse model. We added additional details for the in vivo study to Materials and Methods and included an additional schematic diagram with details of the mouse experiment (Figure 5 A) in the revised Figure 5.

Round 2
